# Cultural & region-specific adaptation of KAP (Knowledge, attitude, and practice) tool to capture healthy lifestyle within primary care settings

**Ahmed Sameer Alnuaimi[1], Muslim Abbas Syed[2]\*, Abduljaleel Abdullatif Zainel[3], Hafiz Ahmed Mohamed[3], Mohamed Iheb Bougmiza[4], Mohamed Ahmed Syed[5]**

1 Department of Clinical Research -Primary Health Care Corporation Qatar, Public Health Research Consultant, Clinical Affairs, Doha, Qatar, 2 Department of Clinical Research -Primary Health Care Corporation Qatar, Research Consultant, Clinical Affairs, Doha, Qatar, 3 Department of Clinical Research -Primary Health Care Corporation Qatar, Public Health Research Consultant, Consultant Community Medicine, Clinical Affairs, Doha, Qatar, 4 Program Director of Community Medicine Residency, Family & Community Residency Program- Primary Health Care Corporation, Doha, Qatar, 5 Department of Clinical Research -Primary Health Care Corporation Qatar, Acting Director of Clinical Research, Clinical Affairs, Doha, Qatar

\* taureanvibes@hotmail.com

**Data Availability Statement:** There are ethical restrictions on sharing data publicly as health data is classified under 'special data'. The Qatari policy

## Abstract

### Background

Non-communicable diseases contribute to a significant global burden of disease and are associated with modifiable risk factors such as physical inactivity, unhealthy diet, tobacco use and excessive alcohol consumption. These risk factors are closely related with lifestyles and eating patterns which are often culturally embedded and managed differently in various health care settings.

### Aim of the study

To assesses the applicability and feasibility of the KAPS (Knowledge, attitude, and practice) survey in generating data about knowledge, attitudes, and practices about healthy lifestyles within eastern Mediterranean settings and providing foundations for testing other models or development of a newer model in this area which captures and influence behavior changes towards healthy lifestyles.

### Methods

The KAP survey was tailored to capture the construct of healthy lifestyles (within the context of Qatar primary care settings) by reviewing existing surveys, adaptation to local context, expert consultation and feedback, pilot testing, feedback analysis, cognitive interviews and translation and validation.

for data protection prohibits sharing data publicly without signing a legal agreement. The datasets used can be issued upon reasonable request by contacting clinical research department, primary healthcare corporation (email to contact is researchsection@phcc.gov.qa).

**Funding:** This study was funded by the Primary Health Care Corporation. The funders had no role in study design, data collection and analysis, decision to publish, or preparation of the manuscript.

**Competing interests:** The authors have read the journal's policy and the authors of this manuscript have the following competing interests to declare: This study was funded by the Primary Health Care Corporation. There are no patents, products in development or marketed products associated with this research to declare. This does not alter our adherence to PLOS ONE policies on sharing data and materials.

**Abbreviations:** KAP, Knowledge, attitude, and practice; NCDs, non-communicable diseases.

## Results

The study reports that most participants found the content comprehensive, relevant, easy to understand but considered it lengthy. Analysis of grading of the 73 questionnaire items (complete questionnaire included as supplementary document) included by the panel of experts (n = 13) demonstrated that more than half of questions (52.1%) have a CVR value of 1. Thematic analysis of overall perceptions of the service users (n = 11) pertaining to the feasibility of the KAP survey identified three important themes which included i) clarity & readability of the questions ii) relevance of the instrument and iii) factors influencing service users' participation in survey.

## Conclusion

A culturally sensitive and region specific KAP tool specifically designed for healthy lifestyles can aid in health advocacy, monitoring modifiable risk factors, capturing rich epidemiological data to design preventive interventions, surveillance of high risks patients and strengthening the existing health information systems. Further research is needed to explore evidence-based methodologies to formulate an age-specific and shorter version of KAPs survey without compromising the validity of the tool within specific regional primary healthcare settings.

## Introduction

The World Health Organization recognizes 'health' as a multifaceted concept [1]. In recent literature there is increased emphasizes on the significance of the wider determinants of health particularly understanding the socio-cultural and economic aspects in which healthcare interventions are implemented [2,3]. Various tools have been designed and utilized to capture such information by conducting cross sectional surveys. The most widely used and known tool in this regard is the knowledge, attitude, and practice (KAP) survey [4–6]. There are various modifiable risk factors associated with non-communicable diseases such as poor diet, physical inactivity or sedentary lifestyles and addiction such as tobacco consumption [7,8]. These modifiable risk factors can be controlled through healthy lifestyle practices and can prevent the occurrence of non-communicable diseases and reduce the overall prevalence [9,10]. Therefore, it is important to define the construct of healthy lifestyles and its significance towards preventing or reducing the prevalence of non-communicable disease within Qatar and its primary health care settings.

Moreover, literature substantiates the fact that healthy lifestyles is an important construct and needs to be captured utilizing models which need to be tailored considering the diverse socio-cultural and economic factors within different geographical settings that can influence it [11,12]. The KAP survey at its inception was utilized in family planning and population studies [13]. The tool was applied to capture information on knowledge, attitudes, and practices in family planning that could further direct program initiatives across the globe. The KAP survey over the last few decades has established as one of the most common methodologies to investigate health behavior and to extract information on health seeking practices. However, despite its widespread application there is critique within the scientific committee pertaining to the applicability of the tool in various geographical settings and the accuracy of the data provided about knowledge, attitudes, and practices which can be utilized for purposes of planning of health program [14]. Moreover, there are other models that may be considered when a

program is designed to fill the gap in knowledge pertaining to specific conditions leading to behavioral changes and favorable associated outcomes. These models include Behavioral Learning Theory, Health Belief Model, Social Cognitive Theory, Theory of Reasoned Action or Theory of Planned Behavior, and Ecological and Social Ecological Models [15].

Non communicable diseases are a major public health concern worldwide and contributes to a significant portion of burden of disease. Alarmingly, Qatar has one of the highest prevalence of diabetes worldwide and a high proportion of people visiting non-communicable disease (NCD) clinics in primary care settings have diabetes [16–19]. To plan an intervention at the national level using primary health care data a multiphase project was set into motion. The first part assessed the magnitude of NCD problem and the non-modifiable risk factors. The current study is the second phase of the project exploring the modifiable risk factor. There is limited evidence pertaining to the socio-cultural and region specific KAP's tool applicability in comprehensively capturing the concept healthy lifestyle within primary care settings in Qatar. The main aim of the study is to assesses the cultural and region-specific applicability and feasibility of the KAPS survey in generating data about knowledge, attitudes, and practices about healthy lifestyles among patients accessing primary care.

The findings of the methodology study provide foundations for testing other models or development of a newer tools which are culturally sensitive & region-specific to capture healthy lifestyles to achieve the desired health outcomes within primary care settings in the state of Qatar and similar healthcare settings globally.

## Methods

The KAP survey was tailored to capture the construct of healthy lifestyles (within the context of Qatar primary care settings) including the following steps:

1. **Review Existing Surveys:** The first step included review of existing KAP surveys related to healthy lifestyles and NCD prevention to identify relevant questions and items to outline the foundations of the survey. A pool of relevant items and domains encompassing knowledge, attitude and practices components were mainly derived from the WHO STEPwise survey instrument [20] and other KAP surveys (utilized in other studies pertaining to non-communicable diseases and its associated risk factors) [21,22].

2. **Adaptation to Local Context:** The draft version of the modified KAP tool was verified and amended to comply with Qatar Dietary Guidelines [23] to assure that the tool is culturally suitable for the local population.

3. **Expert Consultation and feedback:** The first draft of the modified tool was shared by a panel of 13 experts which mainly included two Community Medicine Consultant Professors, two Community Medicine Consultants with research background, one community and lifestyle medicine expert, five health couches (dietitians), two community medicine specialists and one public health expert from 25/02/2023 to 14/03/2023. Verbal consent was obtained from the participants to engage in this exercise. The participants were given detailed information about the purpose of the research and the significance of their feedback in context to promoting healthy lifestyles and preventing non-communicable diseases and were given the opportunity to ask any questions. The participants were then asked directly about their willingness to participate in the study. This step was undertaken to ensure the content validity [24] of the tool. For each questionnaire item, the panelist indicated whether the specific knowledge/ Attitude/ Practice item measured is essential, useful but not essential, or not necessary as a contributor to the overall scale measure of that construct. In addition, the panelist indicated the reason for unfavorable (not essential) rating

and writes any suggestion to improve the phrasing of the question. The reasons for unfavorable opinion on the side of the panelist was coded as 1 = irrelevant to the construct tested, 2 = Too technical and difficult to the audience, 3 = Repeated elsewhere and 4 = others (specify). The study tool underwent iterative amendments and improvements based on the structured panelist review feedback. The content validity ratio (CVR) for each item was computed. To determine whether an item is essential, its minimum value was compared with minimum values that depend on the number of experts who contributed ratings (a value of 0.62 for a panel of 10 experts) [25]. Analysis of CVR values of the original version of the questionnaire form showed that more than 50% of questions had a CVR value of 1 (complete agreement). Another 26% had a value of 0.85 and 16.5% had a CVR of 0.69. Smaller CVR values were obtained in 4 questions only (around 5%).

$$CVR_i = \frac{n_e - \frac{N}{2}}{\frac{N}{2}},$$

Where:
CVR$_i$ = value for an item on the test
n$_e$ = number of experts indicating that an item is essential
N = total number of experts in the panel

4. **Considering cultural sensitivity:** Moreover, the panel also provided feedback regarding the various items to ensure that the survey considered cultural sensitivities and religious beliefs specific to Qatar to eliminate or rephrase any items that might be considered too technical, insensitive, or offensive.

5. **Pilot Testing:** The next step involved administering the initial draft of the survey to a small sample (n = 11) of primary care patients attending healthcare facilities in Qatar from 25/04/2023 to 05/05/2023. Verbal consent was taken from services users before participating in the survey. The participants were given detailed information about the purpose of the research and the significance of their feedback and were given the opportunity to ask any questions. The participants were then asked directly about their willingness to participate in the study. Their feedback on strengthened the face validity [26] of the instrument mainly relating to question clarity, comprehension, and relevance [27,28]. The responses to the open ended question were analysed using framework thematic analysis [29]. This approach encompasses 'interpreting, exploring, and reporting patterns and clusters of meaning within the given data' [30] and was facilitated by reading and re-reading the text of responses to open ended question for a full familiarisation.

6. **Feedback Analysis:** The findings of the pilot testing were utilized to analyze and determine whether there were any problematic questions, confusing wording, or items that were not culturally relevant in the modified KAP tool.

7. **Further modification and Refinement:** Based on the feedback analysis, necessary modifications were made to the survey items.

8. **Cognitive Interviews:** Furthermore, cognitive interviews were conducted with a subset of participants to explore how they interpreted and responded to the modified survey items [31,32]. The interviews were conducted utilizing unstructured questions. A total of 15 participants were recruited for the cognitive interviews. The demographic details of the participants that participated in the cognitive interviews is shown in Table 1. The interviews were conducted individually among participants. This step was included to further identify any

**Table 1. Demographic details of participants for cognitive interviews (n = 15).**

| Profession/ status | n | Age | Gender | Ethnicity | Years of experience/ accessibility to primary health services |
|---|---|---|---|---|---|
| Family Medicine Physicians | 4 | 35–45 years of age | 3, Male, 1 Female | 2, Europe, 2 Eastern Mediterranean | Between 5–10 years |
| Dieticians | 4 | 25–34 years of age | 3, Female, 1 Male | 1, Africa, 2 Eastern Mediterranean, 1, Europe | Less than 5 years |
| Public Health Officers | 3 | 25–34 years of age | 2, Males, 1 Female | 3, Eastern Mediterranean | Between 5–10 years |
| Service users of primary care clinics | 4 | 18–65 years of age | 2, Male, 2 Female | 3, Eastern Mediterranean, 1, East Asia | Between 5–10 years |

underlying misconceptions or confusion pertaining to the culturally adapted and region specific KAP tool. The cognitive interview data was analyzed by synthesis and reduction which is a iterative process [33,34]. This involved the following steps [35]:

i) Generating interview text: This initial step was implemented to generate interview text by collecting narratives from the respondents.

ii) Producing detailed summaries: This step involved synthesis of the interviews into summaries to further produce detailed summaries. The main aim of this step was to describe how and why each participants interpreted the questions in the KAPs survey, and the thought process involved in formulating the responses in context to their experiences with the services and interpretation of health outcomes highlighting the difficulties and challenges in generating these responses.

iii) Producing thematic schema: This step involved identifying and mapping common themes pertaining to the detailed phenomena captured and the processes involved in generating a response.

iv) Advanced schema: This step involved constructing advanced schema by comparing advanced themes across subgroups.

v) Drawing conclusions: The last step involved determining and explaining the performance of a question and its function in context to the socio-cultural background of the participants and their experiences

8. **Finalization of the tool:** The feedback from both pilot testing rounds, and cognitive interviews was consolidated to make final adjustments to the survey items, ensuring they accurately captured participants' knowledge, attitudes, and practices related to healthy lifestyles and NCD prevention.

**Translation and Validation:** The survey was translated into Arabic to maintain the accuracy of the content. A validation process (backward translation) was conducted to ensure the translated version maintained the intended meaning.

## Results

### Demographic details of the expert panel

The demographic details of the expert panel involved in grading each questionnaire item (for each questionnaire item) for the initial modified KAP tool for healthy lifestyles is demonstrated in Table 2. A significant percentage include dieticians as shown in Table 1. The work

**Table 2. Demographic details of the expert panel (n = 13).**

| Job title | Years of experience | N (%) | Gender (Female (n), Male (n)) | Panelists nationality in context to WHO regions (n) |
|---|---|---|---|---|
| Health coaches (dietitians) | Between 2 to 5 years | 5 (38.5) | Female (3), Male (2) | Eastern Mediterranean (3), Europe (2) |
| Community Medicine Consultant Professors | More than 10 years | 2(15.4) | Female (1), Male (1) | Eastern Mediterranean (2) |
| Community Medicine Consultant with research background | Between 5–10 years | 2(15.4) | Male (2) | Eastern Mediterranean (1), Africa (1) |
| Community medicine specialists | Between 5–10 years | 2(15.4) | Male (2) | Eastern Mediterranean (2), |
| Community and lifestyle medicine expert | Between 5–10 years | 1(7.7) | Female (1) | Africa (1) |
| Public health expert | More than 10 years | 1(7.7) | Male (1) | Europe (1) |

experience of dieticians ranged between 2 to 5 years, whereas community medicine consultant professor and public health expert had more than 10 years of experience (Table 1).

### Content validity ratio values (CVR)

Analysis of grading of the 73 questionnaire items (complete questionnaire included as supplementary document) included by the panel of experts (n = 13) demonstrated that more than half of questions (52.1%) have a CVR value of 1 which is followed by 26% with a value of 0.85 as depicted in Table 3.

### Service users' (n = 11) responses of the KAP feasibility survey

Most of the participants (n = 8) took 45–60 minutes to complete the survey and responded that the questions were clear and easy to understand (91%), were relevant to and covered important domains of healthy diet and lifestyles, physical activity and associated attitudes as illustrated in Tables 4–6.

Findings of the thematic analysis of the open-ended question pertaining to service users' perceptions pertaining to the KAPs feasibility survey.

Thematic analysis of the open-ended question about overall perceptions of the service users pertaining to the KAPs feasibility survey identified 3 important themes which were mainly i) clarity & readability of the questions ii) relevance of the instrument and iii) factors influencing service users' participation in survey. The main themes with subsequent sub-themes and related quotes are demonstrated in Table 7.

i) Clarity & readability of the questions

The service users were overall satisfied with the clarity of the questions included in the KAPs feasibility survey. However, it was highlighted that the level of education and overall awareness about the general wellbeing can affect the interpretation and the clarity of the questions.

Similarly, a male Qatari service user commented:

**Table 3. Frequency distribution of the CVR (Content Validity Ratio) of the 73 questionnaire items.**

| CVR Value | N | % |
|---|---|---|
| 1 | 38 | 52.1 |
| 0.7–0.92 | 19 | 26 |
| 0.62–0.69 | 12 | 16.4 |
| <0.62 | 4 | 5.5 |
| **Total** | **73** | **100** |

**Table 4. Describing the results of KAP feasibility survey on a sample of 11 service users: Anticipated duration of completing the questionnaire.**

|  | N | % |
|---|---|---|
| Approximately how long did it take you to complete the questionnaire? |  |  |
| 15 minutes | 0 | 0.0 |
| 30 minutes | 3 | 27.3 |
| 45 minutes | 2 | 18.2 |
| 60 minutes | 6 | 54.5 |
| Total | 11 | 100.0 |
| What is your opinion of the length of the questionnaire in assessing healthy eating and active lifestyle? |  |  |
| Too long | 9 | 81.8 |
| Too short | 0 | 0.0 |
| Appropriate | 2 | 18.2 |
| Total | 11 | 100.0 |

**Table 5. Results of pilot testing of KAP feasibility survey on a sample of 11 service users: General feedback on questionnaire.**

| Positive feedback (Total N = 11) | N | % |
|---|---|---|
| The questions were clear | 10 | 90.9 |
| Found the wording of some questions difficult to understand? | 3 | 27.3 |
| Felt that the questionnaire omits issues considered to be important to investigate | 0 | 0.0 |
| Had difficulties completing the questionnaires | 0 | 0.0 |
| Had additional comments or suggestions for improvement | 0 | 0.0 |

**Table 6. Describing the results of KAP feasibility survey on a sample of 11 service users.**

| Positive feedback (Total N = 11) | N | % |
|---|---|---|
| **Relevance of healthy diet section** |  |  |
| Thought that the questions about the Healthy Diet knowledge section are relevant to measuring knowledge about healthy food and eating? | 10 | 90.9 |
| Thought that the questions about the Healthy Diet attitudes section are relevant to measuring attitudes towards healthy food and eating? | 11 | 100.0 |
| Thought that the questions about the Healthy Diet practice section are relevant to measuring healthy food and eating behaviors? | 11 | 100.0 |
| **Relevance of physical activity section** |  |  |
| Thought that the questions about the Physical activity knowledge section are relevant to measuring knowledge about the role of active lifestyle and healthy weight? | 11 | 100.0 |
| Thought that the questions about the Physical activity attitude section are relevant to measuring attitudes towards active lifestyle and healthy weight? | 11 | 100.0 |
| Thought that the questions about the Physical activity practice section are relevant to measuring the physical activity practice? | 11 | 100.0 |

'The questionnaire reads well overall but not all parts can be the same. There must be few questions which might seem difficult to some. It is easy survey to complete though'.

ii) Relevance of the survey instrument

Regarding the relevance of the survey instrument the services users highlighted the important of the cultural context and relatability of the tool within a local context. The services

**Table 7. Thematic analysis of perceptions of services users pertaining to the KAPS tool.**

| Theme | Sub-theme | Quotes |
|---|---|---|
| Clarity & readability of the questions | Variance in various questions | 'The questionnaire reads well overall but not all parts can be the same. There must be few questions which might seem difficult to some. It is easy survey to complete though'. |
| | Level of understanding | 'Different people have different levels of education and basic understanding about their health. It can affect their interpretation and for some it can very clear what the question is asking but might be not for some'. |
| Relevance of the survey instrument | Capturing what it claims | 'I personally feel the questions in the survey can collect the data about my health and lifestyle choice that influence them. The tool seems to be true to what is made for, in delivering the information it claims to'. |
| | Relatability to lifestyles within local community | 'After completing the survey, I feel that it did cover my lifestyle which can affect my health. I think it does relate to the general population in Qatar.' |
| | Cultural context | 'Every culture is different, how people live eat and maintain their health. A good tool must be made keeping this in mind'. |
| Factors influencing service users' participation in survey | Length of the tool | 'I am usually a very busy person and if I have little time while visiting my doctor, I will not be that keen to fill the survey as I feel it's a bit long and requires time to complete about my lifestyle and other health information.' 'It doesn't bother me much because I realize if I fill this can benefit my health and the community. But I feel it will be different for different patients. How they feel as the survey has a lot of questions, for me I don't mind doing it'. |

users documented the survey to be relevant and to truly captured what it claimed.

For example, an expat service user quoted relating to the significance of cultural context of the survey:

'Every culture is different, how people live eat and maintain their health. A good tool must be made keeping this in mind'.

iii) Factors influencing service users' participation in survey.

The services users highlighted the issue with the overall length of the survey and acknowledge that it can be influenced by individual patients' preferences due to their daily life commitments and schedules during their healthcare visits as described in Table 6.

## Discussion

The key findings of the study provide empirical evidence and application of standardized steps (based on theory and practice) pertaining to tailoring existing tools and surveys to formulate a KAP questionnaire to capture healthy lifestyles in a specific regional and cultural settings. The study highlights the significance of incorporating expert feedback in this exercise, acknowledging that it is an iterative process and recognizing the challenges associated with the items and domains included in such a tool. The study reports that most participants found the content comprehensive, relevant, easy to understand but considered it lengthy.

Evidence suggests that while designing a KAP tool one of the essential principles is to consider the completeness of the questionnaire that it comprehensively captures the knowledge, attitudes and practices relating to the specific key research topic being investigated [36–38]. There are challenges associated with that due to the time constrains within healthcare settings. The healthcare providers are expected to deliver healthcare services within defined time frames which is further exacerbated with additional workload and unexpected patient influx particularly in emergency departments [39–41]. The administration of such a tool can be perceived as an additional task for healthcare providers and patients which makes the length of the questionnaire a key issue. The findings of the study further substantiate the existing evidence that individual patient preferences and daily life commitments can impact the willingness to

participate in the survey. There is a trade-off between the comprehensiveness of the questionnaire and compliance of the services users' participation.

Moreover, healthy lifestyles are a multi-faceted and complex concept and can vary significantly within different geographical settings based on cultural and sociodemographic factors. Our study attempted to tailor the tool according to Qatar dietary guidelines. It is important to consider regional dietary patterns when adopting similar strategies to design such a tool for a specific setting.

As previously mentioned, non-communicable diseases contribute to a significant burden of disease globally and have economic implications due to the high costs associated with managing these disease conditions which are chronic in nature and have adverse health outcomes including a poor quality of life and disability [12,13]. Application of a KAP tool specifically designed for healthy lifestyles can aid in health advocacy, monitoring the modifiable risk factors, capturing rich epidemiological data to design preventive interventions, surveillance of high risks patients and strengthening the existing health information systems.

The key strengths of the study include that it adhered to the key theoretical (evidence based) steps mentioned in literature to design the tool. Feedback was received by panel of experts pertaining to the inclusion of the relevant domains and items within the tool while considering the cultural and religious sensitives. It was ensured to eliminate or rephrase any items that might be considered too technical, insensitive, or offensive content validity and the relevance of the tool for the cohort of participants it is designed for. Moreover, the findings of the pilot testing were utilized to analyze and determine whether there were any problematic questions, confusing wording, or items that were not culturally relevant in the modified KAP tool. One of the limitations of the study is that the tool is region specific and cannot be applied in non-eastern Mediterranean settings. However, this limitation is due to complying to the key principle while designing such a tool that it is modified in accordance with the target population and healthcare systems understanding and preferences towards knowledge, attitudes, and practices to healthy lifestyles. It can also be argued that this strategy further strengthens the tool for capturing healthy lifestyle knowledge in Qatar's primary care setting.

Moreover, it is important to acknowledge that age (particularly elderly) can also play an important role in the willingness of the service users to participate in the KAPs survey. Further research can focus on the elderly population. However, this was beyond the scope of the present study.

## Conclusion

A culturally and region specific designed KAP tool for healthy lifestyles have a broad range of benefits for patients and lead to desired health outcomes within different population groups. Such a tool can aid in health advocacy, monitoring modifiable risk factors, capturing rich epidemiological data to design preventive interventions, surveillance of high risks patients and strengthening the existing health information systems within a specific geographic setting. Further research is needed to explore evidence-based methodologies to formulate an age-specific and shorter version of KAPs survey without compromising the validity of the tool and keeping in view the cultural sensitivity of the tool and the dynamics of specific regional primary healthcare settings.

## Supporting information

**S1 File. Questionnaire (modified KAPS tool to capture healthy lifestyles within primary care settings).**
(DOCX)

## Author Contributions

**Conceptualization:** Ahmed Sameer Alnuaimi, Muslim Abbas Syed, Abduljaleel Abdullatif Zainel, Mohamed Iheb Bougmiza, Mohamed Ahmed Syed.

**Formal analysis:** Muslim Abbas Syed.

**Methodology:** Muslim Abbas Syed, Mohamed Iheb Bougmiza.

**Writing – original draft:** Ahmed Sameer Alnuaimi, Muslim Abbas Syed, Abduljaleel Abdullatif Zainel, Hafiz Ahmed Mohamed, Mohamed Ahmed Syed.

**Writing – review & editing:** Ahmed Sameer Alnuaimi, Muslim Abbas Syed, Abduljaleel Abdullatif Zainel, Hafiz Ahmed Mohamed, Mohamed Iheb Bougmiza, Mohamed Ahmed Syed.

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
