## [Decision Letter · Decision Letter 0]

17 Jun 2024

PONE-D-24-12828Designing a KAP (Knowledge, attitude, and practice) tool to capture healthy lifestyle within primary carePLOS ONE

Dear Dr. Syed,

Thank you for submitting your manuscript to PLOS ONE. After careful consideration, we feel that it has merit but does not fully meet PLOS ONE’s publication criteria as it currently stands. Therefore, we invite you to submit a revised version of the manuscript that addresses the points raised during the review process.

**The reviewers have provided a number of comments for you to consider, and which are designed to help you clarify and strengthen the manuscript. Please consider each comment in your rebuttal and revised manuscript.**

We look forward to receiving your revised manuscript.

Kind regards,

Jenny Wilkinson, PhD

Academic Editor

PLOS ONE

Journal Requirements:

2. In the online submission form, you indicated that The datasets used and/or analyzed during this study are available from the corresponding author on reasonable request.

5. We note that this data set consists of interview transcripts. Can you please confirm that all participants gave consent for interview transcript to be published?

If they DID provide consent for these transcripts to be published, please also confirm that the transcripts do not contain any potentially identifying information (or let us know if the participants consented to having their personal details published and made publicly available). We consider the following details to be identifying information:

- Names, nicknames, and initials

- Age more specific than round numbers

- GPS coordinates, physical addresses, IP addresses, email addresses

- Information in small sample sizes (e.g. 40 students from X class in X year at X university)

- Specific dates (e.g. visit dates, interview dates)

- ID numbers

Or, if the participants DID NOT provide consent for these transcripts to be published:

- Provide a de-identified version of the data or excerpts of interview responses

- Provide information regarding how these transcripts can be accessed by researchers who meet the criteria for access to confidential data, including:

a) the grounds for restriction

b) the name of the ethics committee, Institutional Review Board, or third-party organization that is imposing sharing restrictions on the data

c) a non-author, institutional point of contact that is able to field data access queries, in the interest of maintaining long-term data accessibility.

d) Any relevant data set names, URLs, DOIs, etc. that an independent researcher would need in order to request your minimal data set.

For further information on sharing data that contains sensitive participant information, please see: https://journals.plos.org/plosone/s/data-availability#loc-human-research-participant-data-and-other-sensitive-data

If there are ethical, legal, or third-party restrictions upon your dataset, you must provide all of the following details (https://journals.plos.org/plosone/s/data-availability#loc-acceptable-data-access-restrictions):

a. A complete description of the dataset

b. The nature of the restrictions upon the data (ethical, legal, or owned by a third party) and the reasoning behind them

c. The full name of the body imposing the restrictions upon your dataset (ethics committee, institution, data access committee, etc)

d. If the data are owned by a third party, confirmation of whether the authors received any special privileges in accessing the data that other researchers would not have

e. Direct, non-author contact information (preferably email) for the body imposing the restrictions upon the data, to which data access requests can be sent

Reviewers' comments:

Reviewer's Responses to Questions

**Comments to the Author**

1. Is the manuscript technically sound, and do the data support the conclusions?

Reviewer #1: Partly

Reviewer #2: Yes

2. Has the statistical analysis been performed appropriately and rigorously? 

Reviewer #1: No

Reviewer #2: Yes

3. Have the authors made all data underlying the findings in their manuscript fully available?

Reviewer #1: No

Reviewer #2: Yes

4. Is the manuscript presented in an intelligible fashion and written in standard English?

Reviewer #1: Yes

Reviewer #2: Yes

5. Review Comments to the Author

**Reviewer #1:** The presents study is conducted to assess KAP (Knowledge, attitude, and practice) tool to capture healthy lifestyle within primary care, the study is done one the indigenous sample of Qatar and may benefit from the following suggestions

Abstract

• Background: Briefly mention specific examples of modifiable risk factors (e.g., diet, physical activity)

• Aim: Instead of "globally," specify the target population (e.g., in a specific region or cultural context).

• Conclusion: Emphasize the broader impact beyond modifiable risk factors. Can the KAPS contribute to designing interventions to improve health outcomes?

Consider shortening sentences to improve readability, especially in the background and methods sections.

Introduction

• Shorten the opening about the multifaceted nature of health. Briefly state that various factors influence health outcomes for individuals and communities.

• Combine similar points. Merge the sections about the importance of socio-economic factors and the critique of the KAPS tool.

• Highlight the gap your research addresses. Emphasize the lack of research on the KAPS tool's applicability in capturing healthy lifestyle knowledge in Qatar's primary care setting.

• Cite sources directly related to Qatar and NCDs. Focus on references that highlight the prevalence of NCDs and modifiable risk factors in Qatar.

• State your research question clearly. Briefly state what you aim to achieve by assessing the KAPS tool in this context.

Methods

• Sample Size: While you mention pilot testing with 11 participants, a larger sample size might be recommended to strengthen the generalizability of the pilot testing findings. What is the reference for it?

• Description of Pilot Testing and Cognitive Interviews: Consider elaborating on the format of the pilot testing and cognitive interviews (e.g., individual interviews, focus groups).

• Demographic characteristics of the participants should be added

• Data Analysis for Pilot Testing and Cognitive Interviews: mention how you analyzed the feedback from the pilot testing and cognitive interviews.

• Though it’s a validation study it lacks statistical rigor in terms of validation

The most important question is why the author has chosen this particular method of development validation, are these best practices, if yes please quote references of the same

Results

• Focus on Content Validity: The results section primarily focuses on demographics and completion time. Emphasize the percentage of items with acceptable CVR (e.g., >0.62) to demonstrate content validity.

• Pilot Testing Results: Provide more details on the feedback from service users beyond percentages. Briefly mention specific comments on clarity, relevance, and length of the survey.

• Data on Unfavorable Ratings: If the expert panel provided reasons for unfavorable ratings on CVR (irrelevant,difficult, etc.), consider mentioning the frequency of these reasons.

• Combine Findings: The results section currently describes service user feedback in separate sections for healthy diet and physical activity. Consider combining these sections for improved readability.

Discussion

• Connect Findings to Existing Literature: While you mention the importance of completeness in KAP surveys (citations provided), elaborate on how your study addresses this considering the length concerns raised by participants.

• Address Length Concerns: Acknowledge the trade-off between completeness and length. Discuss potential strategies to shorten the survey while maintaining validity, such as branching logic or creating different versions for specific contexts.

• Generalizability vs. Specificity: The discussion acknowledges limitations in generalizability due to the region-specific nature of the tool. However, emphasize the strengths of the tool for capturing healthy lifestyle knowledge in Qatar's primary care setting.

**Reviewer #2:** 1. KAP was created by making some adjustments to the culture of the local community. What customs in the local community are unique and known to increase the risk of NCDs? Is there any previous research on this?

2. is there a particular reason for involving respondents who are all individuals with a professional background in the health sector? why did the panelists not include people outside the health profession?

3. The KAP is intended to be completed independently by patients, with input from panelists who consider the KAP too long, is the KAP reliable for use in the elderly population?

4. Panelists' characteristics other than professional backgrounds were not explained. Age characteristics may affect their perception of the length or shortness of the questionnaire.

5. What type of questions were used? Can you explain a bit about it, for example, Likert scale, multiple choice, or open question?

6. It has not been explained which part of the KAP tool makes it too long and takes a long time to fill in.

6. PLOS authors have the option to publish the peer review history of their article (what does this mean?). If published, this will include your full peer review and any attached files.

Reviewer #1: No

Reviewer #2: No

---

## [Author Response · Author response to Decision Letter 0]

17 Jul 2024

Dear Editor, 

We would like to thank you and the reviewers for the invaluable comments and feedback on our submitted manuscript, " Designing a KAP (Knowledge, attitude, and practice) tool to capture healthy lifestyle within primary care". We appreciate the time and effort that you and the reviewers have invested in meticulously evaluating our work and the attention to detail to the various components of the study.

We have carefully reviewed the feedback and made a thorough effort to address all the reviewers' comments, revise the manuscript and make the necessary changes accordingly. We believe these revisions have strengthened the manuscript and addressed most of the reviewers' concerns.

We have now submitted a revised manuscript with tracked changes for your review as instructed. We hope that the revisions meet with your approval and look forward to any further feedback you may have.

Thank you again for your kind support.

Sincere Regards

Dr Syed M Abbas 

MBChB MPH MSc MRCGP MCPS HPE MS PhD 

Reviewer #1: The presents study is conducted to assess KAP (Knowledge, attitude, and practice) tool to capture healthy lifestyle within primary care, the study is done one the indigenous sample of Qatar and may benefit from the following suggestions

Abstract

• Background: Briefly mention specific examples of modifiable risk factors (e.g., diet, physical activity)

Response: Thank you for the comment. We have now mentioned some examples of modifiable risk factors in the background section as kindly suggested.

• Aim: Instead of "globally," specify the target population (e.g., in a specific region or cultural context).

Response: Thank you for the important suggestion. We have now changed ‘globally’ to eastern Mediterranean settings which is more specific to study as kindly highlighted in your comment.

• Conclusion: Emphasize the broader impact beyond modifiable risk factors. Can the KAPS contribute to designing interventions to improve health outcomes?

Consider shortening sentences to improve readability, especially in the background and methods sections.

Response: Thank you for highlighting an important aspect of significance of KAP tool beyond modifiable risk factors. The broader aspects are now highlighted in the conclusion (marked in red). The suggestion of shortening sentences has also been carefully considered in the text of the manuscript.

Introduction

• Shorten the opening about the multifaceted nature of health. Briefly state that various factors influence health outcomes for individuals and communities.

Response: Thank you for your kind suggestion. It was insightful suggestion from your end and is much appreciated. It improves the readability and further strengthens the introduction component. The opening of multifaceted nature of health is now shortened by deleting a sentence as highlighted in track changes in the revised main manuscript. The diverse range of social, economic, and environmental factors are now mentioned within the wider determinants of health as kindly suggested.

• Combine similar points. Merge the sections about the importance of socio-economic factors and the critique of the KAPS tool.

Response: Thank you for the important suggestion. The paragraph pertaining to the importance of socio-economic factors and the critique of the KAPS tools is merged as kindly suggested.

• Highlight the gap your research addresses. Emphasize the lack of research on the KAPS tool's applicability in capturing healthy lifestyle knowledge in Qatar's primary care setting.

Response: Thank you for highlighting an important point. This has been incorporated in the introduction section highlighted in track changes. 

• Cite sources directly related to Qatar and NCDs. Focus on references that highlight the prevalence of NCDs and modifiable risk factors in Qatar.

Response: Thank you for highlighting an important point. The references have been added which refer to NCDs, Modifiable risk factors in Qatar. This is highlighted in track changes.

• State your research question clearly. Briefly state what you aim to achieve by assessing the KAPS tool in this context.

Response: Thank you for the important suggestion. The research question of the study is now clearly highlighted in the last paragraph as kindly advised. As suggested, we have now mentioned that the findings of the methodology study provide foundations for testing other models or development of a newer model in this area which can capture and influence behavior changes towards healthy lifestyles and desired health outcomes within primary care settings in the state of Qatar. 

Methods

• Sample Size: While you mention pilot testing with 11 participants, a larger sample size might be recommended to strengthen the generalizability of the pilot testing findings. What is the reference for it?

Response: Thank you for the insightful comment. We truly appreciate the attention to detail from your end to further improve the study. The references are as below and added in text thanks to your kind suggestion:

a. Andrade C, Menon V, Ameen S, Kumar Praharaj S. Designing and Conducting Knowledge, Attitude, and Practice Surveys in Psychiatry: Practical Guidance. Indian J Psychol Med. 2020;42(5):478-481. Published 2020 Aug 27. doi:10.1177/0253717620946111.

b. Guidelines for Assessing Nutrition-related Knowledge, Attitudes and Practices. Rome: Food and Agriculture; 2014 (http://www.fao.org/3/i3545e/i3545e05.pdf, accessed on 07/07/2024)

• Description of Pilot Testing and Cognitive Interviews: Consider elaborating on the format of the pilot testing and cognitive interviews (e.g., individual interviews, focus groups).

Response: Thank you for highlighting an important point to strengthen the methods section. We have now added details on pilot testing and cognitive interviews as kindly suggested. Changes are highlighted in track changes.

‘The interviews were conducted individually among participants. This step was included to further identify any underlying misconceptions or confusion. The cognitive interview data was analyzed by synthesis and reduction(1, 2). This involved the following steps i) the initial step of conducting cognitive interviews to generate interview text (collecting narratives from respondents) ii) synthesizing detailed summaries of interview text iii) producing thematic schema by comparing summarizes across respondents iv) constructing advanced schema by comparing advanced themes across subgroups and v) drawing conclusions’. (3)

• Demographic characteristics of the participants should be added

Response: Thank you for the important suggestion. The demographic details of the panelists are now added in table 1 as kindly suggested (highlighted in track changes).

• Data Analysis for Pilot Testing and Cognitive Interviews: mention how you analyzed the feedback from the pilot testing and cognitive interviews.

Response: Thank you for the insightful comment. As kindly suggested details on analysis of the cognitive interviews are now added. The changes are highlighted as track changes in the main document. The results of pilot testing are depicted in table 4 as frequency and distribution (highlighted in track changes).

‘The cognitive interview data was analyzed by synthesis and reduction(1, 2). This involved the following steps i) the initial step of conducting cognitive interviews to generate interview text (collecting narratives from respondents) ii) synthesizing detailed summaries of interview text iii) producing thematic schema by comparing summarizes across respondents iv) constructing advanced schema by comparing advanced themes across subgroups and v) drawing conclusions’. (3)

The responses to the open ended question were analysed using framework thematic analysis.(4) This approach encompasses ‘interpreting, exploring, and reporting patterns and clusters of meaning within the given data’(5) and was facilitated by reading and re-reading the text of responses to open ended question for a full familiarisation

• Though it’s a validation study it lacks statistical rigor in terms of validation

Response: Thank you for the insightful and important comment. The current study had no intention of testing the psychometric properties of a scale. Instead, it focused on developing a KAP survey that is culturally suitable for the targeted multinational population of Qatar. The objective was to validate the alignment of the study tool with Qatari dietary guidelines. Thanks to your previous suggestion of clearly highlighting the study question in the introduction component this reads more clearly in the manuscript now.

The most important question is why the author has chosen this particular method of development validation, are these best practices, if yes please quote references of the same

Response: Reply: Thank you for the insightful comment. The WHO established principles of a KAP survey design were followed in choosing the current method (ref b). Relying on CVR and the panel of experts for developing the tools was described in ref c.

b. Guidelines for Assessing Nutrition-related Knowledge, Attitudes and Practices. Rome: Food and Agriculture; 2014 (http://www.fao.org/3/i3545e/i3545e05.pdf, accessed on 07/07/2024)

c. Hiew CC, Chin YS, Chan YM, Mohd Nasir MT (2015) Development and Validation of Knowledge, Attitude and Practice on Healthy Lifestyle Questionnaire (KAP-HLQ) for Malaysian Adolescents. J Nutr Health Sci 2(4): 407. doi: 10.15744/2393-9060.2.407 Volume 2 | Issue 4 Journal of Nutrition and Health Sciences

Results

• Focus on Content Validity: The results section primarily focuses on demographics and completion time. Emphasize the percentage of items with acceptable CVR (e.g., >0.62) to demonstrate content validity.

Response: Thank you for your important and insightful point. This information is highlighted in track changes as kindly suggested both in methods and result section of the manuscript.

‘Analysis of CVR values of the original version of the questionnaire form showed that more than 50% of questions had a CVR value of 1 (complete agreement). Another 26% had a value of 0.85 and 16.5% had a CVR of 0.69. Smaller CVR values were obtained in 4 questions only (around 5%)’. 

Where:

CVRi = value for an item on the test 

ne = number of experts indicating that an item is essential

N = total number of experts in the panel

Content validity ratio values (CVR)

Analysis of grading of the 73 questionnaire items (complete questionnaire included as supplementary document) included by the panel of experts (n=13) demonstrated that more than half of questions (52.1%) have a CVR value of which is followed by 26% with a value of 0.85 as depicted in table 2.

Table 2: Frequency distribution of the CVR (Content Validity Ratio) of the 73 questionnaire items.

CVR Value N %

1 38 52.1

0.7-0.92 19 26

0.62-0.69 12 16.4

<0.62 4 5.5

Total 73 100

• Pilot Testing Results: Provide more details on the feedback from service users beyond percentages. Briefly mention specific comments on clarity, relevance, and length of the survey.

Response: Thanks to your important suggestion we did thematic analysis of the responses of the open-ended question about feedback of the service users which is now described additionally in results section and depicted in table 6. We are grateful to your kind suggestion has it has further strengthened the results section of the study and highlights qualitative information from service users’ perspective.

Findings of the thematic analysis of the open-ended question pertaining to service users’ perceptions pertaining to the KAPs feasibility survey

Thematic analysis of the open-ended question about overall perceptions of the service users pertaining to the KAPS tool identified 3 important themes which were mainly i) clarity & readability of the questions ii) relevance of the instrument and iii) factors influencing service users’ participation in the survey. The main themes with subsequent sub-themes and related quotes are depicted in table 6.

i) Clarity & readability of the questions

The service users were overall satisfied with the clarity of the questions included in the KAPs feasibility survey. However, it was highlighted that the level of education and overall awareness about the general well-being can affect the interpretation and the clarity of the questions.

Similarly, a male Qatari service user commented:

The questionnaire reads well overall but not all parts can be the same. There must be few questions which might seem difficult to some. It is easy survey to complete though’.

ii) Relevance of the survey instrument

Regarding the relevance of the survey instrument the services users demonstrated the important of the cultural context and relatability of the tool. The services users documented the survey to be relevant and to truly captured what it claimed. 

For example, an expat service user quoted relating to the significance of cultural context of the survey:

‘Every culture is different, how people live eat and maintain their health. A good tool must be made keeping this in mind’.

iii) Factors influencing service users’ participation in the survey

The services users highlighted the issue with the overall length of the survey and acknowledge that it can be influenced by individual patients’ preferences due to their daily life commitments and schedules during their healthcare visits as shown in table 6.

Table 6 Thematic analysis of perceptions of services users pertaining to the KAPS tool.

Theme Sub-theme Quotes

Clarity & readability of the questions Variance in various questions

 ‘The questionnaire reads well overall but not all parts can be the same. There must be few questions which might seem difficult to some. It is easy survey to complete though’.

 Level of understanding ‘Different people have different levels of education and basic understanding about their health. It can affect their interpretation and for some it can very clear what the question is asking but might be not for some’.

Relevance of the survey instrument Capturing what it claims

 ‘I personally feel the questions in the survey can collect the data about my health and lifestyle choice that influence them. The tool seems to be true to what is made for, in delivering the information it claims to’.

 Relatability to lifestyles within local community

 ‘After completing the survey, I feel that it did cover my lifestyle which can affect my health. I think it does relate to the general population in Qatar.’

 Cultural context

 ‘Every culture is different, how people live eat and maintain their health. A good tool must be made keeping this in mind’.

Factors influencing service users participation in the survey Length of the tool ‘I am usually a very busy person and if I have little time while visiting my doctor, I will not be that keen to fill the survey as I feel it’s a bit long and requires time to complete about my lifestyle and other health information.’

‘It doesn’t bother me much because I realize if I fill it this can benefit my health and the community. But I feel it will be different for different patients. How they feel as the survey has a lot of questions, for me I don’t mind doing it’.

• Data on Unfavorable Ratings: If the expert panel provided reasons for unfavorable ratings on CVR (irrelevant, difficult, etc.), consider mentioning the frequency of these reasons.

Response: Thank you for the insightful comment. This information is highlighted in track changes as kindly suggested.

 Questions that received unfavorable rating and labeled as "not necessary" as a contributor to the overall scale measure of a specific construct by two of the panelists were further examined for the reason of using that label. If the reasons were "irrelevant to the construct tested", or "Repeated elsewhere" this question was deleted. Reporting the reason as being "too technical and difficult for the audience," in addition to the two previously reported ones by any panelist requires amendments to that question. The second round of the panelist validation exercise con

---

## [Decision Letter · Decision Letter 1]

2 Aug 2024

PONE-D-24-12828R1Designing a KAP (Knowledge, attitude, and practice) tool to capture healthy lifestyle within primary carePLOS ONE

Dear Dr. Syed,

Thank you for submitting your manuscript to PLOS ONE. After careful consideration, we feel that it has merit but does not fully meet PLOS ONE’s publication criteria as it currently stands. Therefore, we invite you to submit a revised version of the manuscript that addresses the points raised during the review process.

Your responses and manuscript revisions have been reviewed by the original reviewers and both reviewers have highlighted that while the work is of interest it lacks methodological and analytical rigor. Having reviewed their original and subsequent comments, and your revisions, I invite you to provide future revisions to address their concerns. Please focus on the clarity of the purpose of this study (is it about development of a KAP instrument or the modification of existing tools for a new audience) and the methodological rigor of the study, ensuring all relevant details are provided (e.g. how many individuals were interviewed, on what basis was n=11 deemed valid for the pilot study etc). I also request that you consider the title of your work and whether this may be modified to improve clarity and alignment with your study.

We look forward to receiving your revised manuscript.

Kind regards,

Jenny Wilkinson, PhD

Academic Editor

PLOS ONE

Reviewers' comments:

Reviewer's Responses to Questions

**Comments to the Author**

1. If the authors have adequately addressed your comments raised in a previous round of review and you feel that this manuscript is now acceptable for publication, you may indicate that here to bypass the “Comments to the Author” section, enter your conflict of interest statement in the “Confidential to Editor” section, and submit your "Accept" recommendation.

Reviewer #1: (No Response)

Reviewer #2: (No Response)

2. Is the manuscript technically sound, and do the data support the conclusions?

Reviewer #1: No

Reviewer #2: Partly

3. Has the statistical analysis been performed appropriately and rigorously? 

Reviewer #1: N/A

Reviewer #2: No

4. Have the authors made all data underlying the findings in their manuscript fully available?

Reviewer #1: Yes

Reviewer #2: Yes

5. Is the manuscript presented in an intelligible fashion and written in standard English?

Reviewer #1: Yes

Reviewer #2: Yes

6. Review Comments to the Author

Reviewer #1: 1) The language of the manuscript needs to be refined for redundancies and syntax errors. Some paragraphs in the introduction can be merged together.

2) The title "Designing a KAP (Knowledge, attitude, and practice) tool to capture healthy lifestyle within primary care", should be rewritten to familiarize the reader upfront with what is being done in this study. The word 'designing' is too vague, and connotates with the development of a tool. As I read it, the manuscript is concerned not with the designing of KAP questionnaire, but aligning and modifying it with the cultural specifications. This should be hinted at in the title.

3) The methodology is not sufficiently rigorous.

5) The pilot testing sample is too low.

6) Elaborate more on what exactly was done in the cognitive interview, specifying if the interview was structured or unstructured. If the interview was structured, then what steps were taken.

7) CVI ratings can be disclosed in the result section of the abstract, if feasible.

Reviewer #2: Thank you for responding to the review,

This study is interesting enough to assess KAP (Knowledge, attitude, and practice), so it is expected to capture a healthy lifestyle in primary care. This study certainly raises important issues that can be used for the local community in Qatar. However, this manuscript does not report well enough on how the development of these tools can fit these interests.

Stronger arguments need to be made about the uniqueness of the local community about the NCDS so that these tools are necessary or can help address the problems.

Research methods related to building instruments need to be strengthened. It is necessary to report with clear stages when research is done with mixed methods and appropriate analysis for Likert scales.

But in general, this manuscript is fascinating. Hopefully, it can be improved further. Good luck.

7. PLOS authors have the option to publish the peer review history of their article (what does this mean?). If published, this will include your full peer review and any attached files.

Reviewer #1: No

Reviewer #2: No

---

## [Author Response · Author response to Decision Letter 1]

7 Aug 2024

Response to reviewer comments 

Comment: Your responses and manuscript revisions have been reviewed by the original reviewers and both reviewers have highlighted that while the work is of interest it lacks methodological and analytical rigor. Having reviewed their original and subsequent comments, and your revisions, I invite you to provide future revisions to address their concerns. Please focus on the clarity of the purpose of this study (is it about development of a KAP instrument or the modification of existing tools for a new audience) and the methodological rigor of the study, ensuring all relevant details are provided (e.g. how many individuals were interviewed, on what basis was n=11 deemed valid for the pilot study etc.). I also request that you consider the title of your work and whether this may be modified to improve clarity and alignment with your study.

Response: Thank you for the feedback, Jenny. We have carefully considered the points that you have mentioned and reviewer comments whilst submitting the revised version of the manuscript. We have modified the title of the study to provide a clear indication what they study has to offer. We have added more details in the methodology section pertaining to cognitive interviews. We have also highlighted the analysis details of the quantitative and qualitative data that was collected during the pilot testing phase of the study. The study highlights both the expert panel feedback (CVR) and pilot testing of the KAPs survey to capture service users perspective/feedback (both quantitatively and qualitatively). We agree that the pilot testing sample is not high. However, cognitive interviews (n=14) and quantitative & qualitative data was collected from service users during the pilot testing (n=11) of phase of the study provide a comprehensive perspective of the cultural and region-specific adaptation of the tool both from healthcare professionals and service users. Moreover, the tools are also adapted with expert panel feedback (n=13) in the early steps of the study.

Response to reviewer 1 &2 comments

Reviewer #1: 

1) The language of the manuscript needs to be refined for redundancies and syntax errors. Some paragraphs in the introduction can be merged.

Response: Thank you for your comment. We have re-read the manuscript and merged paragraphs in the introduction section as kindly advised. Moreover, we have reviewed the manuscript to correct redundancies or any syntax errors. Thank you for suggesting.

2) The title "Designing a KAP (Knowledge, attitude, and practice) tool to capture healthy lifestyle within primary care", should be rewritten to familiarize the reader upfront with what is being done in this study. The word 'designing' is too vague, and connotates with the development of a tool. As I read it, the manuscript is concerned not with the designing of KAP questionnaire but aligning and modifying it with the cultural specifications. This should be hinted at in the title.

Response: Thank you for your kind suggestion. As advised, we have revised the topic of the study and now it reads as, ‘Cultural & region-specific adaptation of KAP (Knowledge, attitude, and practice) tool to capture healthy lifestyle within primary care settings".

3) The methodology is not sufficiently rigorous.

Response: Thank you for the comment. As kindly suggested, we have added details in the cognitive interviews, describing each step. We have now added details of analysis of both the quantitative and qualitative data that was collected during the pilot phase in this methodology study. 

5) The pilot testing sample is too low.

Response: Thank you for the comment. We agree that the pilot testing sample is not high. However, cognitive interviews (n=14) and quantitative & qualitative data was collected from service users during the pilot testing (n=11) of phase of the study provide a comprehensive perspective of the cultural and region-specific adaptation of the tool both from healthcare professionals and service users. Moreover, the tools are also adapted with expert panel feedback (n=13) in the early steps of the study.

6) Elaborate more on what exactly was done in the cognitive interview, specifying if the interview was structured or unstructured. If the interview was structured, then what steps were taken.

Response: Thank you for the suggestion. We have now provided details in the methods section (highlighted as track changes) of each step of the cognitive interviews as well as data analysis of the qualitative data analyzed during the pilot testing of the KAPs survey. 

* Cognitive Interviews: Cognitive interviews were conducted with a subset of participants to explore how they interpreted and responded to the modified survey items(1, 2). The interviews were conducted utilizing unstructured questions. A total of 15 participants were recruited for the cognitive interviews. The demographic details of the participants that participated in the cognitive interviews is shown in table 1. The interviews were conducted individually among participants. This step was included to further identify any underlying misconceptions or confusion pertaining to the culturally adapted and region specific KAP tool. The cognitive interview data was analyzed by synthesis and reduction which is a iterative process(3, 4). This involved the following steps(5):

 i) Generating interview text: This initial step was implemented to generate interview text by collecting narratives from the respondents. 

ii) Producing detailed summaries: This step involved synthesis of the interviews into summaries to further produce detailed summaries. The main aim of this step was to describe how and why each participants interpreted the questions in the KAPs survey, and the thought process involved in formulating the responses in context to their experiences with the services and interpretation of health outcomes highlighting the difficulties and challenges in generating these responses. 

iii) Producing thematic schema: This step involved identifying and mapping common themes pertaining to the detailed phenomena captured and the processes involved in generating a response. 

iv) Advanced schema: This step involved constructing advanced schema by comparing advanced themes across subgroups. 

v) Drawing conclusions: The last step involved determining and explaining the performance of a question and its function in context to the socio-cultural background of the participants and their experiences 

Table 1 Demographic details of participants for cognitive interviews (n=15)

Profession/ status n Age Gender Ethnicity Years of experience/ accessibility to primary health services

 Family Medicine Physicians 4 35- 45 years of age 3, Male, 1 Female 2, Europe, 2 Eastern Mediterranean Between 5-10 years

 Dieticians 4 25-34 years of age 3, Female, 1 Male 1, Africa, 2 Eastern Mediterranean, 1, Europe Less than 5 years

Public Health Officers 3 25-34 years of age 2, Males, 1 Female 3, Eastern Mediterranean Between 5-10 years

Service users of PHCC 4 18-65 years of age 2, Male, 2 Female 3, Eastern Mediterranean, 1, East Asia Between 5-10 years

7) CVI ratings can be disclosed in the result section of the abstract, if feasible.

Response: Thank you for the important point. We have now included CVI rating in the results section of the abstract as kindly suggested. We have also highlighted the main themes that resulted from the cognitive interviews in the results section of the abstract. 

Reviewer #2

 Thank you for responding to the review, this study is interesting enough to assess KAP (Knowledge, attitude, and practice), so it is expected to capture a healthy lifestyle in primary care. This study certainly raises important issues that can be used for the local community in Qatar. However, this manuscript does not report well enough on how the development of these tools can fit these interests. Stronger arguments need to be made about the uniqueness of the local community about the NCDS so that these tools are necessary or can help address the problems.

Response: We have carefully considered the points that you have mentioned and reviewer comments whilst submitting the revised version of the manuscript. We have modified the title of the study to provide a clear indication what they study has to offer. We have also included the points that you have highlighted in the introduction and discussion component.

Research methods related to building instruments need to be strengthened. It is necessary to report with clear stages when research is done with mixed methods and appropriate analysis for Likert scales.

Response: As kindly suggested, we have added details in the cognitive interviews, describing each step. Moreover, details are highlighted in red of the analysis steps of both the quantitative and qualitative data that was collected in this methodology study. The methods section details 9 steps in total overall constituting more than 1000 words of the manuscript.

But in general, this manuscript is fascinating. Hopefully, it can be improved further. Good luck.

Response: Thank you for your kind words and encouragement. It means a lot.

1. Bekerian D, Dennett J. The cognitive interview technique: Reviving the issues. Applied Cognitive Psychology. 1993;7(4):275-97.

2. Memon A, Gawrylowicz J. The cognitive interview. The handbook of communication skills. 2018:511-30.

3. Miles MB, Huberman AM. Qualitative data analysis: An expanded sourcebook: sage; 1994.

4. Suter B. Tales to transit: Sub-Saharan African migrants’ experiences in Istanbul: Linköping University Electronic Press; 2012.

5. Malterud K. Systematic text condensation: a strategy for qualitative analysis. Scandinavian journal of public health. 2012;40(8):795-805.

---

## [Decision Letter · Decision Letter 2]

15 Oct 2024

Cultural & region-specific adaptation of KAP (Knowledge, attitude, and practice) tool to capture healthy lifestyle within primary care settings

PONE-D-24-12828R2

Dear Dr. Syed,

We’re pleased to inform you that your manuscript has been judged scientifically suitable for publication and will be formally accepted for publication once it meets all outstanding technical requirements.

Kind regards,

Jenny Wilkinson, PhD

Academic Editor

PLOS ONE

Additional Editor Comments (optional):

Thank you for your revisions and responses to reviewer comments; these have satisfactorily addressed reviewer concerns.

Reviewers' comments:

Reviewer's Responses to Questions

**Comments to the Author**

1. If the authors have adequately addressed your comments raised in a previous round of review and you feel that this manuscript is now acceptable for publication, you may indicate that here to bypass the “Comments to the Author” section, enter your conflict of interest statement in the “Confidential to Editor” section, and submit your "Accept" recommendation.

Reviewer #2: All comments have been addressed

2. Is the manuscript technically sound, and do the data support the conclusions?

Reviewer #2: Yes

3. Has the statistical analysis been performed appropriately and rigorously? 

Reviewer #2: Yes

4. Have the authors made all data underlying the findings in their manuscript fully available?

Reviewer #2: Yes

5. Is the manuscript presented in an intelligible fashion and written in standard English?

Reviewer #2: Yes

6. Review Comments to the Author

Reviewer #2: The manuscript has been written in more detail and is clearly and easily read.

There are no further suggestions regarding improvements to the manuscript

7. PLOS authors have the option to publish the peer review history of their article (what does this mean?). If published, this will include your full peer review and any attached files.

Reviewer #2: No

---

## [Editor Report · Acceptance letter]

11 Dec 2024

PONE-D-24-12828R2 

PLOS ONE

Dear Dr. Syed, 

I'm pleased to inform you that your manuscript has been deemed suitable for publication in PLOS ONE. Congratulations! Your manuscript is now being handed over to our production team.

Kind regards, 

on behalf of

Dr Jenny Wilkinson 

Academic Editor

PLOS ONE